# Validation of the Rosenberg Self-Esteem Scale among the Iranian adult population: A cross-sectional study

Saeed Ghasempour[1], Hamid Sharif-Nia[2,3], Soheil Nouri[1],
Seyedmohammad Mirhosseini[4,5], Ali Abbasi[4,6]*

1 Student Research Committee, School of Nursing and Midwifery, Shahroud University of Medical Sciences, Shahroud, Iran, 2 Psychosomatic Research Center, Mazandaran University of Medical Sciences, Sari, Iran, 3 Department of Nursing, Amol Faculty of Nursing and Midwifery, Mazandaran University of Medical Sciences, Sari, Iran, 4 Department of Nursing, School of Nursing and Midwifery, Shahroud University of Medical Sciences, Shahroud, Iran, 5 Student Research Committee, School of Nursing and Midwifery, Guilan University of Medical Sciences, Rasht, Iran, 6 Center for Health Related Social and Behavioral Sciences Research, Shahroud University of Medical Sciences, Shahroud, Iran

* abbasi_msn@yahoo.com

## Abstract

### Background

Self-esteem refers to an individual's overall sense of self-worth, which plays a crucial role in their well-being. One of the most commonly used instruments to measure this concept is the Rosenberg Self-Esteem Scale (RSES), which has been translated and psychometrically validated in numerous cultures and languages to date. Therefore, this study aimed to assess the psychometric properties of the Persian version of the RSES in the Iranian adult population.

### Methods

This cross-sectional validation study was conducted following the Strengthening the Reporting of Observational Studies in Epidemiology (STROBE) guidelines in Shahroud city, northeastern Iran. A total of 533 adults from this city who spoke Persian and were literate were included in the study using convenience sampling. After translating and culturally adapting the RSES in accordance with World Health Organization (WHO) guidelines, face and content validity were assessed using both qualitative and quantitative methods. Additionally, construct validity was evaluated through exploratory and confirmatory factor analysis. Reliability was also determined by calculating Cronbach's alpha coefficient, McDonald's omega coefficient, composite reliability (CR), and intraclass correlation coefficient (ICC).

**Data availability statement:** The data supporting the results of this study contain potentially personally identifying and sensitive information about the participants. Therefore, this data cannot be made publicly available according to the policies and regulations of the Research Ethics Committee of Shahroud University of Medical Sciences. The ethical approval of this study also explicitly prohibits public sharing of the data to protect the confidentiality and privacy of the participants. However, the data will be available upon reasonable request from the corresponding author via email (abbasi_msn@yahoo.com) or from the Vice President for Research and Technology of this university via email (vcr@shmu.ac.ir) to researchers who meet the criteria for access to confidential data.

**Funding:** The author(s) received no specific funding for this work.

**Competing interests:** The authors have declared that no competing interests exist.

## Results

The face and content validity of all items were confirmed through both qualitative and quantitative methods. Based on the results of the exploratory factor analysis using polychoric correlations and robust weighted least squares (WLSMV) estimation, the Persian version of this scale consists of two factors: (1) Positive self-esteem and (2) Negative self-esteem, which account for 59.1% of the total variance of the scale. All goodness-of-fit indices in the confirmatory factor analysis also supported this model (CMIN/$df$ = 2.29, RMSEA = .049, CFI = .981). These two factors showed acceptable internal consistency and stability, as evidenced by Cronbach's alpha coefficients (.828 and .801), McDonald's omega coefficients (.833 and .810), composite reliabilities (.820 and .796), and intraclass correlation coefficients (.838 and .875).

## Conclusion

The findings of the current study showed favorable psychometric properties of the Persian version of the RSES for measuring self-esteem in the Iranian adult population.

## Introduction

Self-esteem is defined as a sense of self-worth or self-respect that plays a key role in an individual's well-being [1,2]. Coopersmith (1990) defined self-esteem as an evaluation that an individual makes of themselves and typically maintains, reflecting an attitude of approval or disapproval [3]. Rosenberg (1965) defined self-esteem as an individual's overall sense of self-worth, derived from an evaluation of one's own characteristics, which can be either negative or positive [4]. Negative self-esteem (NSE) is a risk factor for psychiatric disorders and social problems, leading to self-loathing and pessimism [5,6]. In contrast, individuals with positive self-esteem (PSE) view themselves as worthy of love, respect, and attention. Therefore, it is considered a protective factor for mental health [7–9]. Assessing self-esteem in adults is crucial due to its association with various positive outcomes. High self-esteem in adulthood is linked to increased life satisfaction, reduced likelihood of seeking treatment for physical or mental disorders, improved self-rated mental health, and fewer symptoms of depression or anxiety [10]. Self-esteem plays a central role in the spiritual life and moral behavior of the average Iranian. For many Iranians, humanity is regarded as the crown of creation, and each individual is uniquely valued by God [11]. In Iran, spirituality is positively linked to self-esteem, and it is believed that self-esteem is one of the mechanisms through which spirituality enhances mental health [12]. As a result, maintaining and strengthening self-esteem is considered a cultural value in Iran, even before assessing its potential deficiencies [13]. However, accurately measuring self-esteem in adults requires a valid and reliable instrument.

Common instruments for measuring self-esteem include the Rosenberg Self-Esteem Scale (RSES) [4] and the Coopersmith Self-Esteem Inventory (CSEI) [3]. The RSES was initially created to gauge adolescents' feelings of acceptance or overall self-worth,

while the CSEI was designed to evaluate attitudes towards themselves in specific contexts like parents, peers, school, and personal interests. However, the RSES is a simpler and shorter instrument with superior validity and reliability compared to the CSEI. It can be used with individuals of any age group as long as they have at least a fifth-grade education level [14,15].

This ten-item scale is considered the most widely used instrument for measuring individuals' self-esteem. It has been translated and standardized in numerous societies worldwide [6,7,16–19]. Kielkiewicz et al. (2020) evaluated the construct validity and dimensional structure of the RSES in an Irish sample. Their results indicated that a two-factor model provided a superior fit for the self-esteem items compared with the original one-factor model, with all fit indices showing improvement. The model demonstrated acceptable psychometric values ($\chi^2 = 276.99$, $df = 131$, RMSEA = .06, CFI = .94, IFI = .94), and the AIC further supported the two-factor structure as the better-fitting solution [17]. In a related study, Tulachan et al. (2022) examined the validity and reliability of the RSES in a Nepalese adult population. Principal axis factoring and parallel analysis confirmed a two-factor structure. The scale yielded a Cronbach's alpha coefficient of .711, and item-level test–retest analysis showed that 70% of items had moderate reliability [18]. Likewise, Mayordomo et al. (2020) adapted and validated the RSES for a Spanish older adult population, adhering to the original single-factor model proposed by the scale's developer. Their findings demonstrated good model fit ($\chi^2 = 217.20$, $p < .05$, CFI = .965, GFI = .980, RMSEA = .070) and satisfactory internal consistency (Cronbach's alpha coefficient = .732) [16].

According to its developer, this scale was initially considered a single-factor scale [4]. However, it has been recognized as having both positive and negative factors in recent years. Researchers have used factor analysis to demonstrate that the RSES is actually a two-factor construct consisting of positive and negative self-concepts. Specifically, five items with positive wording (e.g., "On the whole, I am satisfied with myself.") were grouped into one factor, PSE, while five items with negative wording (e.g., "All in all, I am inclined to feel that I am a failure.") were grouped into another factor, NSE [17,20,21]. Several studies have also shown that a single-factor model of this scale does not fit well using factor analysis. Instead, they have found that a two-factor model with positive and negative self-concepts provides a better fit [22,23].

In Iran, studies have also been conducted on the psychometric properties of the Persian version of this scale. Rajabi and Bohlol (2007) conducted a study to assess the validity and reliability of the Persian version of the RSES in first-year dormitory students at Shahid Chamran University [15]. Despite the small sample size, factor analysis was only based on exploratory factor analysis (EFA) in this study, and construct validity was not assessed using confirmatory factor analysis (CFA). However, the statistical population and cultural framework had changed compared to the initial study of instrument design.

Additionally, Rajabi and Karjo Ksmai (2012) conducted a study to examine the confirmatory structure of the two-factor model of the Persian version of RSES [24]. In this study, despite studying the scale in a new community, factor analysis was not conducted with an exploratory approach and relied only on a confirmatory approach. Accordingly, a review of the literature on the evaluation of the psychometric properties of the Persian version of the RSES identified several limitations and shortcomings in the psychometric processes of the existing Persian versions. Most research in this area has examined the validity and reliability of the scale among Iranian students. While students are indeed adults, the broader general adult population encompasses a much wider range of individuals with diverse demographic characteristics. Therefore, developing and validating a version of the scale specifically for the Iranian adult population is essential. Self-esteem is a concept deeply influenced by cultural and contextual factors. To ensure that a measure of such subjective concepts is suitable for a new cultural context, a careful and systematic process of translation and adaptation must be undertaken. So, the present study was designed and implemented to evaluate the validity and reliability of the Persian version of the RSES in Iranian adults.

## Materials and methods

### Design and purpose

This cross-sectional validation study was conducted following the Strengthening the Reporting of Observational Studies in Epidemiology (STROBE) guidelines with the aim of determining the psychometric properties of the Persian version of the RSES in the Iranian adult population. The study took place in Shahroud city, northeastern Iran characterized by its

culturally diverse, predominantly Muslim, Persian-speaking population from February 19, 2024, to August 10, 2024. This setting provided an opportunity to evaluate the Persian version of the Rosenberg Self-Esteem Scale in an adult population with a wide range of demographic characteristics.

## Participants

Although sample size is crucial in factor analysis, different guidelines have been provided for its estimation [25]. One set of guidelines estimates the sample size based on the ratio of N (number of items) to P (number of participants). This ratio varies from 1:3–1:20 in various literature [26,27]. Therefore, since the RSES has only 10 items, the sample size will range between 30 and 200. Another set of guidelines considers a sample size of 100 as poor, 200 as moderate, 300 as good, and 500 as very good [28]. A larger sample size implies fewer measurement errors, more stable factor loadings, and more repeatable factors [29]. Therefore, over 500 eligible Iranian adults were recruited using convenience sampling. This method was selected for its ease of implementation, high questionnaire response rate, and frequent use in comparable studies [30]. Participants were eligible if they were 18–65 years old, fluent in Persian, possessed at least basic literacy skills (sufficient to complete the scale), and had access to online social media platforms. Individuals were excluded if they were using neuroleptic medications or had experienced acute stress within the six months preceding the study. Eligibility was determined based on participants' self-reports. The researchers conducted an online survey by sharing a link to the online data collection form on online social media platforms (such as Telegram, WhatsApp, Eitaa and, e.g.,) and various public information channels throughout the city. Invitations to participate in the study were distributed through channels dedicated to local events and content in Shahroud city. To ensure that participants were residents of Shahroud city, individuals were asked to confirm their place of residence via a self-report question before accessing the online questionnaire. The online form consists of demographic variables (included gender, marital status, educational level, employment status, and age) and the RSES items. According to the database records, 540 participants completed the online form. However, seven respondents were excluded from the analysis because they did not complete the RSES-specific section. As a result, the effective response rate was estimated at 98.7%. These details are presented in the study's flowchart (Fig 1).

## Rosenberg Self-Esteem Scale (RSES)

This scale, designed by Rosenberg in 1965, measures adolescents' feelings of acceptance or overall self-worth. According to its developer, the RSES comprises 10 items that form a single-factor scale [4]. However, recent studies using factor analysis have suggested that this scale is a two-factor construct consisting of PSE (items 1, 2, 4, 6, and 7) and NSE (items 3, 5, 8, 9, and 10) [17,20,21]. The criteria for scoring items are based on a four-point scale, with "strongly agree" receiving a score of 4, "agree" receiving a score of 3, "disagree" receiving a score of 2, and "strongly disagree" receiving a score of 1. Items 3, 5, 8, 9, and 10 are also scored in reverse order [4].

## Translation and cultural adaptation

The institution that owns this scale was initially contacted via e-mail to request permission for its translation and psychometric evaluation in Persian. Subsequently, the translation and cultural adaptation were conducted following the guidelines of the World Health Organization (WHO) [31], which involved the following steps:

**1. Forward translation in Persian.** Initially, this scale was independently translated by two main translators ($T_1$ and $T_2$) who were fully fluent in both English and Persian language and culture. The translators were instructed to avoid literal translations while remaining true to the original English text. Additionally, during the translation process, they noted all suitable equivalents for English phrases or terms, which could be referenced in later stages and replaced if necessary. As a result, two distinct Persian translations of the RSES were produced.

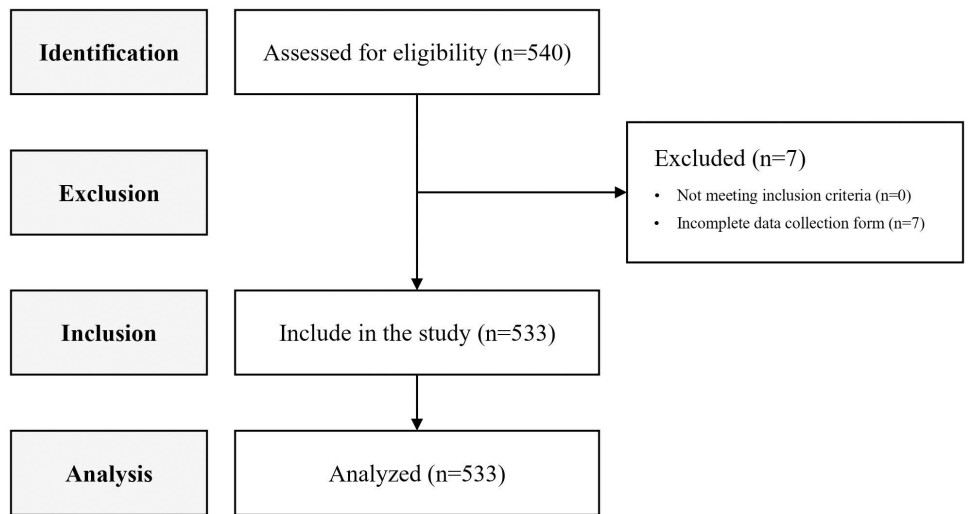

**Fig 1. STROBE flowchart of study.**

**2. Analysis and synthesis of early Persian.** A team consisting of two clinical psychologists (CP$_1$ and CP$_2$) and the main translators (T$_1$ and T$_2$) reviewed the two separate existing translations and their corresponding Persian equivalents. The team evaluated both translations, discussed the differences, and resolved any discrepancies. Finally, after considering all options for equivalence of phrases or terms, a single Persian version of the scale was prepared.

**3. Backward translation into English.** Subsequently, two additional translators (T$_3$ and T$_4$) who were fluent in both English and Persian language and culture and were not previously informed about the English version of the scale or the procedures of the current study translated the Persian version into the source language or English. As a result, two distinct English translations were also obtained from the RSES.

**4. Analysis and composition of secondary English translations.** A team consisting of two clinical psychologists (CP$_1$ and CP$_2$), the main translators (T$_1$ and T$_2$), and the first author (SG) collaborated to evaluate the two existing English translations and create a cohesive English version of the scale. This unified version was then compared to the original English version of the scale. Ultimately, the revised version was sent via email to the institution that owns the scale for approval [32,33].

**5 Expert committee evaluation.** An expert committee, comprised of two clinical psychologists (CP$_1$ and CP$_2$), all translators (T$_1$, T$_2$, T$_3$, and T$_4$), and some of the authors of this article (SG and AA), reviewed the combined Persian forward-translated version (obtained in stage two) and the combined English backward-translated version (obtained in stage four). They assessed whether specific words or phrases conveyed the same ideas or themes in both societies to ensure the accuracy of the translation of the items, as well as their relevance to the new culture and context [34–36].

**6. Pre-test implementation.** Cognitive interviews were finally conducted with 30 Iranian adults representing a diverse range of demographic information to evaluate the items in terms of comprehension, acceptability, and emotional impact. Through this process, any confusing or misleading items were identified and modified with the assistance of the participants themselves [34,36]. This resulted in the initial form of the Persian version of the RSES, which will undergo further analysis to determine its validity and reliability among Iranian adults.

## Face validity

This study evaluated face validity using both qualitative and quantitative approaches. In the qualitative approach, cognitive interviews were conducted with 10 participants to assess the clarity and difficulty of each item. In the quantitative approach, a

different group of 10 participants rated the importance of each item in measuring self-esteem on a five-point Likert scale. An impact score was calculated for each item by multiplying the percentage of participants who rated it as important (4 or 5) by the item's mean importance rating. Items with an impact score above 1.5 were deemed important and kept [37].

## Content validity

Content validity was assessed by two expert panels consisting of 12 experts, using both qualitative and quantitative approaches. The qualitative assessment of content validity was carried out by experts from both panels, focusing on grammar, appropriate word usage, correct placement, scoring accuracy, and completion time [38].

The first expert panel included six clinical psychologists, two psychiatrists, two psychiatric nurses, and two instrument development experts, totaling 12 experts. In addition to qualitatively assessing content validity, they also evaluated the necessity of each item based on a three-point Likert scale: "necessary=3", "useful but unnecessary=2", and "unnecessary=1". Consequently, the content validity ratio (CVR) was calculated using the following formula:

$$\mathbf{CVR} = \frac{\mathbf{nE} - \frac{\mathbf{N}}{2}}{\frac{\mathbf{N}}{2}}$$

(1)

nE represents the number of experts who consider an item essential, while N is the total number of experts evaluating that item. Lawshe's table outlines the minimum CVR value required based on the number of experts. With each item being assessed by 12 experts, the minimum CVR value was established at .56. Consequently, items with a CVR value exceeding .56 were deemed essential and were retained [39,40].

The second expert panel consisted of three clinical psychologists, three psychiatrists, three psychiatric nurses, and three instrument development experts, totaling 12 experts. In addition to qualitatively evaluating content validity, they also assessed the relevance of each item based on a four-point scale: "not relevant=1", "somewhat relevant=2", "relevant but needs revision=3", and "completely relevant=4". Subsequently, the item-level content validity index (I-CVI) was calculated using the following formula:

$$\mathbf{I-CVI} = \frac{\text{The number of experts who rated this item 3 and 4}.}{\text{The total number of experts}.}$$

(2)

Items with an I-CVI of less than .70 are considered unacceptable and should be removed. Items with an I-CVI between .70 and .79 are questionable and require revision. Items with an I-CVI greater than .79 are considered acceptable and should remain unchanged [41,42].

In addition to the I-CVI, the modified Kappa statistic (MKS) was also calculated to minimize the possibility of chance agreement between experts. MKS was determined using the following formulas:

$$\mathbf{MKS} = \frac{\mathbf{I-CVI} - \mathbf{P_c}}{\mathbf{1} - \mathbf{P_c}} \quad \mathbf{P_c} = \left[\frac{\mathbf{N!}}{\mathbf{A!\,(N-A)!}}\right] \times \mathbf{0.5^N}$$

(3)

N represents the total number of raters, and A represents the number of raters who give the item a score of 3 or 4. Therefore, MKS is considered poor if less than 60%, good if between 60 and 74%, and excellent if more than 74% [43].

## Construct validity

The construct validity of this scale was assessed using factor analysis with two approaches: exploratory and confirmatory. To achieve this goal, the entire dataset of 533 Iranian adults was randomly divided into two independent sets using the

random sample selection feature in SPSS version 26, which employs a simple random sampling method without replacement. The first independent dataset of 270 Iranian adults was used to conduct EFA, and the second independent dataset of 263 Iranian adults was used to conduct CFA.

Demographic equivalence between the two subsamples was verified using independent t-tests for age and chi-square tests for categorical variables (gender, marital status, education level, employment status). No significant differences were found between the two groups (all $p > .05$), confirming their comparability.

Given the ordinal nature of the Likert-type data, appropriate statistical methods were employed. For both EFA and CFA, we used estimation methods suitable for ordinal data. To ensure robustness, we conducted a sensitivity analysis comparing the robust weighted least squares (WLSMV) estimator (appropriate for ordinal data) with the maximum likelihood (ML) estimator (which assumes continuous data). The results confirmed that the factor structure and fit indices remained consistent across both estimation methods, supporting the stability of our findings.

Polychoric correlations were calculated as the input matrix for factor analysis. For EFA, we used WLSMV estimation with Promax rotation, which is recommended for ordinal data. For CFA, we similarly employed the WLSMV estimator in JASP 0.19.3.0, which is appropriate for ordinal and non-normal data.

This approach aligns with recommendations for rigorous psychometric evaluation, as it ensures that the factor structure derived from EFA is cross-validated in a separate sample during CFA, minimizing overfitting and enhancing generalizability [44,45]. Such separation is particularly critical when establishing a new cultural adaptation of a scale, as it strengthens confidence in the stability of the identified dimensions.

The assumption of underlying multivariate normality for ordinal data was examined using appropriate measures. While traditional skewness and kurtosis thresholds are less meaningful for ordinal data, we confirmed that the polychoric correlation matrix was positive definite, indicating no serious violations of the underlying normality assumption. For outlier detection in ordinal data, we employed a combination of methods suitable for categorical variables. We conducted response pattern analysis to identify inconsistent or atypical response patterns across the 10 items. Additionally, we used the mean and standard deviation of each item to flag cases with extreme values (e.g., all items rated as 1 or all as 4). Through this process, 7 cases (1.3% of the total sample) were identified as outliers based on their atypical response patterns and were excluded from subsequent analyses. A sensitivity analysis was conducted by comparing the factor structure with and without these outliers, which showed no substantial differences in factor loadings or model fit indices, confirming the robustness of our result [46].

Kaiser-Meyer-Olkin (KMO) measure of sampling adequacy and Bartlett's test of sphericity were utilized to evaluate the suitability of the data for factor analysis. A significant result for Bartlett's test of sphericity ($p < .01$) indicated that KMO values above .9 were excellent [38]. Furthermore, a critical value (CV) of .3 was employed for item retention, calculated using the following formula:

$$CV = \frac{5.152}{\sqrt{n-2}}$$

(4)

In the formula above, n represents the sample size [47]. It is important to note that items with a communality of less than .2 and a factor loading of less than .3 were eliminated. Furthermore, each factor must contain at least three items [48].

The factor structure obtained through EFA was evaluated using CFA with WLSMV estimation in subsequent steps. Common fit indices appropriate for ordinal data were calculated, including root mean square error of approximation (RMSEA) [RMSEA < .08], comparative fit index (CFI) [CFI > .9], parsimony comparative fit index (PCFI) (adjusted form of CFI) [PCFI > .5], normed fit index (NFI) [NFI > .9], parsimony normed fit index (PNFI) (adjusted form of NFI) [PNFI > .5], goodness of fit index (GFI) [GFI > .9], adjusted goodness of fit index (AGFI) [AGFI > .9], incremental fit index (IFI) [IFI > .9], Tucker-Lewis index (TLI) [TLI > .9], and the ratio of Chi-square to degrees of freedom (CMIN/df) [CMIN/df < 3], for the obtained factor structure [42,47].

## Convergent and discriminant validity

In order to assess convergent and discriminant validity, the method proposed by Fornell and Larcker (1981) was utilized. Firstly, three indices were calculated: maximum shared squared variance (MSV), average variance extracted (AVE), and composite reliability (CR). Generally, AVE values greater than .5, CR values greater than .7, or CR values greater than AVE indicate desirable convergent validity of the scale [49]. It is important to acknowledge that the Fornell-Larcker criterion has contextual limitations. Henseler et al. (2015) demonstrated that this method, along with the evaluation of cross-loadings, often fails to reliably detect discriminant validity issues in typical research scenarios. To address these shortcomings, they proposed the Heterotrait-Monotrait ratio (HTMT) correlation matrix, which is based on the multitrait-multimethod matrix framework [50]. Monte Carlo simulations confirmed that the HTMT approach performs more effectively than traditional methods such as the Fornell-Larcker criterion and (partial) cross-loading analyses [51,52]. Accordingly, this study employed the HTMT method to assess discriminant validity, adopting the standard cutoff value of .85, indicating that values below this threshold reflect distinct constructs [51]. So, the discriminant validity of the RSES was evaluated using a new approach called the HTMT matrix [50].

## Reliability

Finally, the internal consistency and construct reliability of this scale were evaluated by calculating Cronbach's alpha coefficient, McDonald's omega coefficient, maximal reliability (MaxR), and CR of each factor. Values greater than .7 for these indices indicate good internal consistency and construct reliability of this scale [53,54]. Thirty Iranian adults were conveniently selected to complete the scale twice, with a two-week interval between administrations, to assess its stability. The sample was chosen to ensure diverse representation in terms of gender, education, occupation, age, and marital status. The intraclass correlation coefficient (ICC) was then calculated using the test-retest method, employing a mixed two-way effects model. ICC values exceeding .8 are considered desirable [55].

## Rosenberg Self-Esteem Scale (RSES) scores

Finally, the mean (SD) scores for overall self-esteem, PSE, and NSE of participating adults were reported.

## Ethical considerations

The Research Ethics Committee of Shahroud University of Medical Sciences approved the current study with the ethics code IR.SHMU.REC.1402.150. Furthermore, the study objectives were clearly communicated to all participants, and their written informed consent was obtained via an online form prior to the completion of the main forms. Participants were informed of the contents of the informed consent form through an online form, and by checking the confirmation box, they indicated their consent to participate in the study. The authors also adhered to the principles of the Committee on Publication Ethics (COPE) when publishing their findings.

# Results

In this study, almost half of the participants were male (49.7%). Participants' ages ranged from 18 to 65 years, with a mean age of 27.37 (*SD* = 9.82). Additional demographic characteristics of the participants are presented in Table 1, categorized by exploratory and confirmatory factor analyses.

## Face validity

All items demonstrated the necessary clarity, relevance, and importance during the qualitative face validity stage. Furthermore, during the quantitative face validity stage, the impact score of all items exceeded 1.5.

Table 1. Demographic characteristics of participants for exploratory and confirmatory factor analysis, as well as total number of participants.

| Variables | | Total samples (N = 533) | EFA sample (N = 270) | CFA sample (N = 263) |
|---|---|---|---|---|
| | | Frequency (%) | Frequency (%) | Frequency (%) |
| Gender | Male | 265 (49.72) | 117 (21.95) | 148 (27.77) |
| | Female | 268 (50.28) | 153 (28.71) | 115 (21.58) |
| Marital status | Single | 373 (69.98) | 252 (47.28) | 121 (22.70) |
| | Married | 144 (27.02) | 16 (3.00) | 128 (24.02) |
| | Divorced | 9 (1.69) | 2 (.38) | 7 (1.31) |
| | Widowed | 7 (1.31) | 0 (.00) | 7 (1.31) |
| Education level | Primary | 51 (9.57) | 18 (3.38) | 33 (6.19) |
| | Diploma | 182 (34.15) | 112 (21.01) | 70 (13.13) |
| | Associate | 47 (8.82) | 22 (4.13) | 25 (4.69) |
| | Bachelor's degree | 175 (32.83) | 77 (14.45) | 98 (18.39) |
| | Master's degree | 59 (11.07) | 31 (5.82) | 28 (5.25) |
| | Doctorate | 19 (3.56) | 10 (1.88) | 9 (1.69) |
| Employment status | Self-employed | 136 (25.52) | 44 (8.26) | 92 (17.26) |
| | Retired | 9 (1.69) | 1 (.19) | 8 (1.50) |
| | Employed | 91 (17.07) | 33 (6.19) | 58 (10.88) |
| | Unemployed | 69 (12.95) | 11 (2.06) | 58 (10.88) |
| | Student | 228 (42.78) | 181 (33.96) | 47 (8.82) |
| | | Mean (SD) | Mean (SD) | Mean (SD) |
| Age | | 27.37 (9.82) | 22.12 (4.83) | 32.76 (10.89) |

EFA: Exploratory Factor Analysis; CFA: Confirmatory Factor Analysis; %: Percent; SD: Standard Deviation.

## Content validity

Based on suggestions from the expert panel during the qualitative content validity stage, items 1, 2, 3, 4, and 6 were slightly modified in terms of grammar and word usage. However, the CVR values for these items ranged from .83 to 1, and their I-CVI values ranged from .91 to 1. Since the cutoff points for CVR and I-CVI were .56 and .79, respectively, no items were revised or eliminated during the quantitative content validity stage. Therefore, all items were retained for subsequent analyses. Additionally, the MKS values for all items were excellent and above the threshold.

## Item analysis

Prior to factor analysis, a comprehensive item analysis was conducted to examine the psychometric properties of the RSES items (S1–S10). The mean total score for the scale was 28.4 (SD = 4.2), with individual item means ranging from 1.8 (Item S8: "I wish I could have more respect for myself") to 3.1 (Item S1: "I feel that I am a person of worth"). All items demonstrated acceptable variability (SD range = .9–1.2). Item-total correlations ranged from .42 (Item S8) to .76 (Item S2: "I feel I have good qualities"), exceeding the recommended threshold of .30 for scale homogeneity [56]. Cronbach's alpha analysis indicated that no item deletion would substantially improve reliability (α-if-item-deleted range = .82–.86), support-ing the retention of all items for subsequent analyses. These findings suggest that all items meaningfully contributed to the assessment of self-esteem.

## Construct validity

The Kaiser–Meyer–Olkin (KMO) measure of sampling adequacy (.902) and Bartlett's test of sphericity ($\chi^2 = 2161.535$, $df = 45$, $p < .001$) indicated that the sample was suitable for factor analysis. Exploratory factor analysis (EFA) using

polychoric correlations and weighted least squares means and variance adjusted (WLSMV) estimation with Promax rotation revealed two factors with eigenvalues greater than one, which together accounted for 59.1% of the total variance of the scale (PSE and NSE) (see Table 2).

Although item S8 demonstrated a factor loading of .48 and a communality of .25, it was retained due to its theoretical relevance to the negative self-esteem (NSE) factor and its contribution to improving the overall model fit. Following the guidelines of Hair et al. (2019), items with factor loadings ≥ .30 were considered acceptable when supported by strong validity indices and satisfactory internal consistency (Cronbach's α > .80) [57].

Confirmatory factor analysis (CFA) results indicated that all goodness-of-fit indices supported the adequacy of the final model: $\chi^2 = 48.41$, $p < .001$, CMIN/$df = 2.29$, RMSEA = .049, 90% CI [.034, .064], CFI = .981, PCFI = .697, NFI = .966, PNFI = .687, GFI = .973, AGFI = .954, IFI = .981, and TLI = .973. The final model of the Persian version of the RSES is presented in Fig 2.

**Table 2. Exploratory factor analysis with polychoric correlations and robust weighted least squares (WLSMV) estimation on two factors of the Persian version of the RSES (N = 270).**

| Factors | Qn. item | Factor loading | h² | λ | % Variance |
|---|---|---|---|---|---|
| PSE | 2- I feel that I have good qualities. | .993 | .547 | 3.019 | 30.2 |
| | 1- I feel that I am a valuable person. (At least the same value as others) | .858 | .582 | | |
| | 6- I have a positive attitude towards myself. | .621 | .606 | | |
| | 7- I am generally satisfied with myself. | .586 | .522 | | |
| | 4- I can do things as well as other people. | .578 | .324 | | |
| NSE | 9- Sometimes I feel useless. | .907 | .671 | 2.889 | 28.9 |
| | 10- Sometimes I think that I have no skills in any fields. | .834 | .586 | | |
| | 3- All things considered, I usually feel like a failure. | .727 | .547 | | |
| | 5- I feel like I don't have much to be proud of. | .528 | .447 | | |
| | 8- I wish I could have more respect for myself. | .475 | .254 | | |

PSE: Positive self-esteem; NSE: Negative self-esteem; h²: Communalities; λ: Eigenvalue.

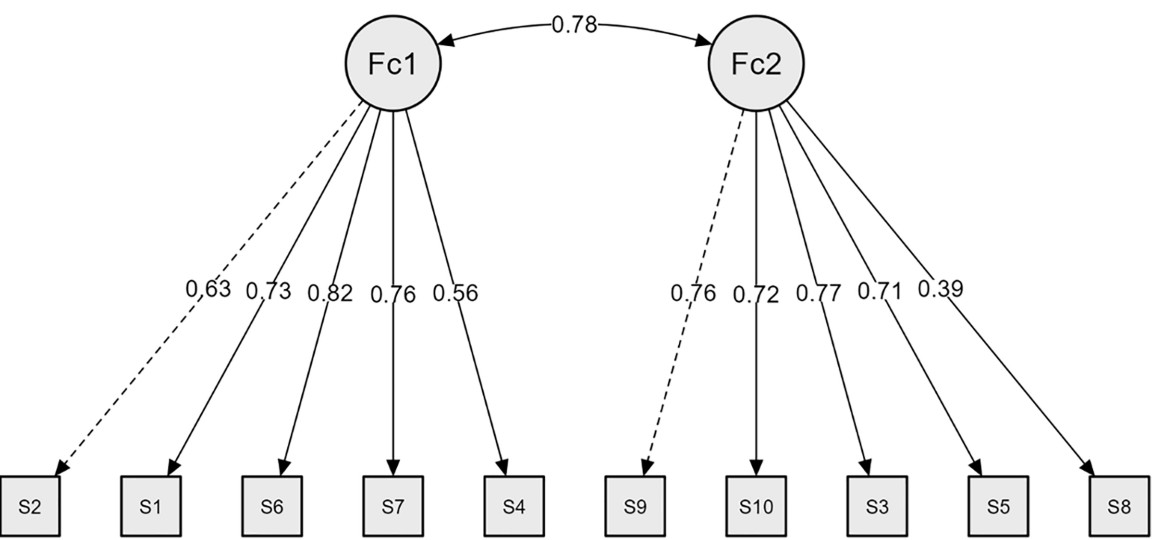

**Fig 2. The final model of the Persian version of the RSES (N = 263).** Fc1: Positive self-esteem; Fc2: Negative self-esteem.

We also tested a one-factor model of the RSES using confirmatory factor analysis on the same sample (N = 263) to allow for a direct comparison of model fit. The one-factor model exhibited poor fit to the data: $\chi^2 = 301.146$, $p < .001$, CMIN/$df$ = 8.604, RMSEA = .120, 90% CI [.107, .132], CFI = .875, PCFI = .681, NFI = .862, PNFI = .670, GFI = .879, AGFI = .810, IFI = .876, and TLI = .840. All fit indices for the one-factor model were substantially worse than those of the two-factor model, supporting the superiority of the two-factor structure.

### Convergent and discriminant validity

As presented in Table 3, the AVE values for both identified factors were slightly below the recommended benchmark of .5. Despite this, the CR values exceeded .7, with CR > AVE for both factors, supporting convergent validity in accordance with Fornell and Larcker (1981). Standardized factor loadings were also calculated, all of which were statistically significant and greater than .5, providing further evidence of convergent validity. The HTMT ratio was .769, indicating an acceptable level of discriminant validity between the two factors. Although this value approaches the critical threshold, it still supports adequate discriminant validity.

### Reliability

As presented in Table 3, the Cronbach's alpha coefficient, McDonald's omega coefficient, MaxR, and CR for each factor were all reported to be above .7, indicating good internal consistency and construct reliability. Meanwhile, the Cronbach's alpha coefficient and McDonald's omega coefficient for the entire scale were .871 and .875, respectively. Furthermore, the ICC for two dimensions of PSE (.838), NSE (.875), and the entire scale (.869) demonstrated acceptable stability.

### Rosenberg Self-Esteem Scale (RSES) scores

The mean scores for adult participants' overall self-esteem, positive self-esteem (PSE), and negative self-esteem (NSE) were 29.87 (SD = 5.51), 16.05 (SD = 2.73), and 13.81 (SD = 3.36), respectively.

### Discussion

The present study aimed to determine the psychometric properties of the Persian version of the RSES among the Iranian adult population. EFA extracted two factors of PSE and NSE from the Persian version of this scale. The first factor, named PSE, consisted of five items and explained 30.2% of the total variance of the scale. The second factor, named NSE, also consisted of five items and explained 28.9% of the total variance of the scale. Together, these two factors accounted for 59.1% of the total variance of the scale.

In this regard, studies conducted to evaluate the psychometric properties of the Persian version of this scale also considered the RSES to be a two-dimensional construct [15,24,41]. For example, the study by Abbasi et al. (2024), which aimed to evaluate the psychometric properties of the Persian version of the RSES in medical sciences students, confirmed the two-dimensionality of this scale. Accordingly, the RSES in Iranian medical sciences students included two factors, PSE (five items) and NSE (five items), which explained a total of 50.7% of the total variance of the scale [41]. In a

**Table 3. Convergent and discriminant validity and reliability of the Persian version of the RSES.**

| Indices factors | CR | AVE | MSV | MaxR (H) | α | Ω | ICC (95% CI) |
|---|---|---|---|---|---|---|---|
| PSE | .820 | .481 | .675 | .843 | .828 | .833 | .838 (.659, .923) |
| NSE | .796 | .448 | .675 | .823 | .801 | .810 | .875 (.736, .940) |

PSE: Positive self-esteem; NSE: Negative self-esteem; CR: Composite Reliability; AVE: Average Variance Extracted; MSV: Maximum Shared Squared Variance; MaxR (H): Maximum Reliability; α: Cronbach's alpha coefficient; Ω: McDonald's omega coefficient; ICC: Intraclass Correlation Coefficient; CI: Confidence Interval.

study by Rajabi and Bohlol (2007), the validity and reliability of the Persian version of this scale were evaluated in first-year dormitory students at Shahid Chamran University. The Persian version of this scale was condensed into two factors: "personal competence" (six items) and "self-satisfaction" (four items), which collectively accounted for 53.8% of the total variance of the scale [15].

This is despite the fact that studies conducted in other societies have presented contradictory findings about the factor structure of this scale [58]. For example, a study by Moksnes et al. (2024) aimed to determine the psychometric properties of the RSES in Norwegian adolescents, showing a two-factor structure for this scale. Specifically, five positively worded items reflected "self-value", and five negatively worded items reflected "self-competence" ($\chi^2/df = 3.07$, $p < .0001$, RMSEA = .055, SRMR = .034, CFI = .98, TLI = .97) [19]. In contrast, Mayordomo et al. (2020) conducted a study to adapt and validate the RSES in the Spanish older adult population based on the single-factor model proposed by the original designer of the scale. After removing items 3, 5, 6, 8, and 9, the single-factor structure of this scale showed favorable validity and reliability in Spanish older adults ($\chi^2 = 217.20$, $p < .05$, CFI = .965, GFI = .980, RMSEA = .070) [16]. It should be noted that the limited studies that considered the RSES to be a unidimensional construct often achieved this desirable structure by eliminating numerous items [7,16]. Therefore, despite the initial designer's statements that the original version of the RSES was unifactorial [4], consistent with most studies in this field, the current study viewed the RSES as a bi-dimensional construct encompassing both PSE and NSE. The variation between the original scale's unidimensional structure [4] and the two-factor structure observed in the Persian version may stem from the application of different statistical techniques and cultural differences between the two populations. The divergence between previous studies—some of which have reported a single-factor structure—and the present study, which identified a two-factor structure, may be attributed to several factors. First, the RSES comprises both positively worded (five items) and negatively worded (five items) statements. Prior research has consistently demonstrated that such wording differences can lead items to cluster into separate factors. This separation does not necessarily reflect distinct underlying constructs, but may instead arise from response style effects or differences in the cognitive processing of positive versus negative statements [59,60]. In Persian, subtle variations in sentence structure, negation, and emphasis may further accentuate this division. Second, although the RSES underwent rigorous forward–backward translation, certain Persian terms for self-worth, competence, or satisfaction may carry implicit connotations absent in the English original. Moreover, the positive and negative items often employ different semantic or idiomatic forms, potentially introducing differential item functioning (DIF) and contributing to the emergence of two factors. Third, cultural context may also play a role. In collectivist or interdependent cultures, such as Iran, the concept of self-esteem may differ from that in Western contexts. In the Iranian cultural setting, factors such as modesty bias and social desirability exert a significant influence on how self-esteem is perceived [61,62]. Positive self-evaluations are often expressed cautiously to avoid perceptions of arrogance, whereas self-critical statements may be more socially acceptable [63,64]. This cultural tendency could produce distinct response patterns, thereby reinforcing the two-factor solution observed in the present study.

In line with the recommendation by Golino et al. (2021) that fit indices should be interpreted relative to alternative models rather than relying solely on absolute cut-off values, we directly compared the fit of a one-factor model with that of the two-factor model [65]. The one-factor model demonstrated poor fit across all indices, whereas the two-factor model showed significantly better fit. This comparison further reinforces the robustness and superiority of the two-factor structure (positive and negative self-esteem) in the Iranian adult population, consistent with the growing body of cross-cultural evidence supporting a bidimensional interpretation of the RSES.

Furthermore, the CFA confirmed the acceptable fit of the proposed model. In this regard, Rajabi and Karjo Ksmai (2012) conducted a study comparing the confirmatory structure of the single-factor and two-factor models of the Persian version of the RSES. The results showed that the two-factor structure of this scale ($\chi^2 = 43.27$, $df = 32$, $p = .088$, $\chi^2/df = 1.35$, RMSEA = .05, CFI = .97, NFI = .90, GFI = .94) had a more favorable fit than its single-factor structure ($\chi^2 = 87.62$, $df = 35$, P = .0001, $\chi^2/df = 2.50$, RMSEA = .10, CFI = .86, NFI = .80, GFI = .88) [24]. In addition, Billa et al. (2023) examined the

psychometric properties of the Telugu version of this scale, which is the fourth most widely spoken language in India. Among the six models proposed, the unidimensional structure of the Telugu version of the RSES, representing perceived positive self-esteem, demonstrated the best fit to the data ($\chi^2 = 4.90$, $df = 5$, $\chi^2/df = .98$, RMR $= .01$, GFI $= .97$, IFI $= .99$, CFI $= .99$, RMSEA $= .0001$) [7]. Although item S8 ('I wish I could have more respect for myself') exhibited a factor loading of .475, it was retained in the final model due to its theoretical relevance to the NSE factor and acceptable communality ($h^2 = .254$). Following recommendations by Hair et al. (2019), items with loadings $\geq .30$ were considered meaningful if they contributed to model fit and scale consistency, particularly when supported by strong overall validity indices (RMSEA $< .08$, CFI $> .90$) [57].

Despite the contradictions found, what distinguishes these two studies from others in this field is that the assessment of construct validity and the implementation of factor analysis are based solely on a confirmatory approach. It can be said that the evaluation of construct validity through factor analysis, using both exploratory and confirmatory approaches, is an integral part of the psychometric process of an instrument. Therefore, the construct validity of the Persian version of the RSES was assessed using both approaches in the present study.

According to Fornell and Larcker's rule (1981), the Persian version of this scale demonstrated favorable convergent validity. Additionally, the HTMT matrix indicated acceptable discriminant validity for the Persian version of the RSES. It is important to note that Rosenberg (1965) did not evaluate the original version of the RSES in this manner [4]. In contrast, Abbasi et al. (2024) utilized the method recommended by Fornell and Larcker (1981) to evaluate the convergent and discriminant validity of the Persian version of this scale. Consequently, CR values above .7, in addition to CR values surpassing AVE, demonstrated its convergent validity. Conversely, MSV values exceeding AVE also indicated its lack of discriminant validity [41]. Despite the consistent issues mentioned earlier regarding the obtained results, the present study confirmed the discriminant validity of the Persian version of the RSES using a newer approach known as the HTMT matrix. In addition, Syropoulou et al. (2021) also demonstrated that the Greek version of this scale exhibits good convergent and discriminant validity using the method proposed by Fornell and Larcker (1981). This was determined by calculating three indices: MSV, AVE, and CR [20]. Therefore, the RSES can be considered an instrument with strong convergent and discriminant validity.

In this study, the internal consistency and construct reliability of each factor were assessed using Cronbach's alpha coefficient, McDonald's omega coefficient, MaxR, and CR. The reported values were all greater than .7. Similarly, in a study by Moksnes et al. (2024), Cronbach's alpha coefficient and CR values for the positive dimension (self-value) and negative dimension (self-competence) of the Norwegian version of this scale were also greater than .8 [19]. Furthermore, a study by Syropoulou et al. (2021) calculated the ordinal alpha and CR values of the three latent variables of general self-esteem (GSE), PSE, and NSE in the Greek version of the RSES to be greater than .8 [20]. This suggests that the RSES exhibits good internal consistency and construct reliability across different cultures and languages.

The stability of the Persian version of this scale was determined using the test-retest method with the help of ICC. The findings showed favorable repeatability of this scale in the Iranian adult population, with the ICC of the entire scale, PSE, and NSE all being more than .8. Similarly, Jiang et al. (2023) achieved an ICC of over .7 for the entire scale and its positive and negative dimensions over a one-week period using a similar approach [6]. Abbasi et al. (2024) also employed the test-retest method to evaluate the stability of this scale. In their study, 40 participants completed RSES in two stages, two weeks apart. As a result, the ICC for the positive and negative dimensions of this scale was reported to be over .8 [41]. Based on this, different studies have also indicated the high stability of RSES, which is consistent with the findings of the current study.

The Persian version of the self-esteem scale includes 10 items assessing two dimensions: positive self-esteem (PSE; items 1, 2, 4, 6, and 7) and negative self-esteem (NSE; items 3, 5, 8, 9, and 10). Each item is rated on a four-point scale ranging from 1 ("I completely disagree") to 4 ("I completely agree"). The negatively worded items (3, 5, 8, 9, and 10) were reverse-coded before computing the total score. A total self-esteem score can be obtained by summing all items after

reverse-coding, with higher scores indicating greater overall self-esteem. Researchers may also calculate separate sub-scale scores for PSE and NSE when a more differentiated assessment of self-esteem dimensions is desired.

## Limitations, strengths, and recommendations

The most significant limitation of this study is the use of a self-report scale, which raises concerns about response bias. A considerable proportion of the participants in this study were young, single students from Shahroud city in northeastern Iran (urban bias), which may restrict the generalizability of the findings to the wider population. Therefore, it is suggested that the mentioned scale be adapted to the culture of other Persian-speaking ethnic groups. While the Persian version of the RSES showed satisfactory psychometric properties, it is essential to recognize some limitations in the cultural adaptation process. Effective cultural adaptation involves more than precise translation; it requires consideration of deeper cultural, social, and psychological factors that influence how respondents understand and express self-esteem. In the context of Iranian culture, dominant social norms may shape individuals' responses, potentially impacting the scale's validity. To address these challenges, future studies should employ more thorough cultural adaptation methods, including qualitative research and collaboration with cultural experts, to better capture the complex and nuanced nature of self-esteem in Iran and improve the scale's relevance and accuracy. This study employed convenience sampling through online forms. While cost-effective and easy to implement, this approach may limit the generalizability of the findings and does not guarantee that participants were in suitable emotional or cognitive states to take part. Future research should take these limitations into account. Although the present study evaluated convergent and discriminant validity using the Fornell–Larcker criterion and the HTMT coefficient, it is recommended that future research further examine convergent validity by assessing correlations between the RSES and theoretically related constructs such as general self-esteem, self-efficacy, quality of life, or well-being. For discriminant validity, future studies may also explore associations with conceptually distinct constructs such as depressive or anxiety symptoms. Such investigations would contribute to a more comprehensive understanding of the construct validity of the two self-esteem dimensions identified in the present study.

Considering these limitations, the present study provided a short, concise, valid, and reliable instrument to measure self-esteem in Iranian adults. This was achieved by employing a novel and robust methodology that addressed the limitations and shortcomings of studies conducted in this field as thoroughly as possible.

## Conclusions

The results of this study demonstrate that the Persian version of the RSES has good validity and reliability for measuring self-esteem in Iranian adults. However, caution should be taken when applying it to married and non-student adults, as the majority of participants in this study were single and student adults.

## Supporting information

**S1 Appendix. Persian version of the Rosenberg Self-Esteem Scale (RSES).**
(DOCX)

## Acknowledgments

This study is the result of a research project approved under number 14020057 at Shahroud University of Medical Sciences. The authors would like to express their gratitude for the support of the Vice Chancellor for Research and Technology at this university. The researchers also sincerely thank all participants and others who collaborated in conducting this study.

## Author contributions

**Conceptualization:** Saeed Ghasempour, Hamid Sharif-Nia, Seyedmohammad Mirhosseini, Ali Abbasi.

**Formal analysis:** Hamid Sharif-Nia.

**Investigation:** Saeed Ghasempour, Soheil Nouri, Seyedmohammad Mirhosseini.

**Methodology:** Hamid Sharif-Nia, Seyedmohammad Mirhosseini, Ali Abbasi.

**Project administration:** Ali Abbasi.

**Supervision:** Ali Abbasi.

**Writing – original draft:** Saeed Ghasempour, Hamid Sharif-Nia, Soheil Nouri, Seyedmohammad Mirhosseini, Ali Abbasi.

**Writing – review & editing:** Saeed Ghasempour, Hamid Sharif-Nia, Seyedmohammad Mirhosseini, Ali Abbasi.

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
