## [Decision Letter · Decision Letter 0]

23 May 2025

Dear Dr. Mirhosseini,

We look forward to receiving your revised manuscript.

Kind regards,

Mohamed Ahmed Said, Ph.D.

Academic Editor

PLOS ONE

Journal Requirements:

Additional Editor Comments:

This study examines the psychometric properties of the Persian Rosenberg Self-Esteem Scale (RSES) in Iranian adults, revealing a stable two-factor structure (PSE/NSE). However, several significant methodological concerns diminish its reliability:

1- Restricted Generalizability: The sample predominantly consists of young, single students, failing to represent the diverse population of Iran.

2- Deficiencies in factor analysis: EFA and CFA likely utilized the same sample, potentially resulting in overfitting, and items with low AVE values (<0.5) lack convergent validity.

3- Insufficient Discriminant Validity: MSV exceeds AVE; yet, HTMT is below 0.85 to support this assertion.

4- Doubtful Item Retention: Certain items exhibit low communalities (<0.2) and loadings (<0.3). More stringent cutoffs (such as >0.4) would be required.

5- Excessive reliance on fit indices: Although the model fit appears satisfactory, the significant χ² value (P<0.001) indicates potential inadequacies in the model's fit.

6- Cross-Cultural Issues: The RSES is typically one-dimensional; however, the Persian version comprises two components: PSE and NSE. Does prejudice exist in culture or translation? Not examined.

7- Reporting Deficiencies: There is an absence of conflict of interest disclosure and no sample division for EFA/CFA validation.

Conclusion: The Persian RSES appears promising; nonetheless, sample bias, issues with factor analysis, and inadequate validity metrics undermine its reliability.

Reviewers' comments:

Reviewer's Responses to Questions

**Comments to the Author**

1. Is the manuscript technically sound, and do the data support the conclusions?

Reviewer #1: Yes

Reviewer #2: Yes

2. Has the statistical analysis been performed appropriately and rigorously?

Reviewer #1: Yes

Reviewer #2: Yes

3. Have the authors made all data underlying the findings in their manuscript fully available?

Reviewer #1: Yes

Reviewer #2: Yes

4. Is the manuscript presented in an intelligible fashion and written in standard English?

Reviewer #1: Yes

Reviewer #2: Yes

Reviewer #1: Comments to the Author

Thank you for the opportunity to review the manuscript, which is about Validity and reliability of the Rosenberg Self-Esteem Scale: A methodological study among the Iranian adult population

The content is very good buy I recommend some suggestions in order to improve it.

First, the paper should be revised by a native English speaker to refine the grammatical problems

1. Please attach the translation permission from the original author of the instrument.

2. Given that the scale was previously developed, why were face and content validity done quantitatively? What if the results recommended deleting item(s) before conducting a factor analysis? Is there any possibility to delete the item(s)? In this case, what were the benefits of conducting these time-consuming processes?

3. Please explain why an Exploratory Factor Analysis (EFA) was conducted on a previously developed scale with well-determined factors. In such cases, as a validation process, conducting a Confirmatory Factor Analysis (CFA) to ensure the model's fitness is sufficient. Please justify the reason for conducting an EFA before a CFA. What was unclear that needed to be "explored"?

4. In the construct validity section, specify exactly how many samples were included in the study for each of the stages of exploratory and confirmatory construct validity?

5. Before conducting exploratory factor analysis, item analysis is performed to examine the correlation between each item's score and the scores of other items and the total score. Additionally, items that affect reliability are identified. Based on this section, the mean and standard deviation for the entire questionnaire and each item (including the highest and lowest scores) are reported. These details have not been reported here.

6. Based on the suggestions of the expert panel during the qualitative content validity stage, some items were revised and modified…(Face and content validity..P10)...

Please specify which items were modified and in what way during the content validity process.

Were the items that were rewritten or modified during content validity adjustment changed because they did not achieve an adequate CVI score, or was there another reason for the changes?

7. In the results section, please mention the names of the extracted factors and indicate whether these items were the same in the original version and the translated version. Were their arrangement and order different or the same? This should also be explained in the discussion section.

8. . To mention what countries did this questionnaire

9. What is the reason for doing psychometrics in Iran.

10. In Table 1, the samples should be written separately. (EFA and CFA)

11. Why are the samples in separate exploration and confirmation steps with reference and reference؟

12. The discussion could be expanded to address potential limitations in the cultural adaptation process. For example, how cultural differences might influence responses beyond translation accuracy.

Reviewer #2: thanks for inviting me to review this study. here there are some recommendations for authors :

Why measuring self-esteem in adults is important?

What is the problem with self-esteem in your country?

Your reasons for conducting a new study were not convincing. Instead of emphasizing whether or not confirmatory factor analysis was conducted, you should focus on the study population because most of the studies were on students.

Move the study population section to the beginning of the method after “Design and purpose”.

Reference 21 should be corrected; the link is not active.

Since nowadays many studies are published in the field of psychometrics and readers are familiar with the basic concepts, it would be better to write the translation section, face, and content validity, more concisely.

what was the Data collection procedure? Online? Paper? what were the demographic variables? Where do you find the samples? what was the response rate?

How do you select 30 Iranian adults for ICC?

please report gender in the text of results, also the range for age

What was the method of keeping items in CFA? Because you keep item S8 with 0.4 in CFA.

Please provide 95%CI for reliability indices.

Please provide a table for the Heterotrait-monotrait ratio of correlations (HTMT)

**Do you want your identity to be public for this peer review?** For information about this choice, including consent withdrawal, please see our Privacy Policy

Reviewer #1: No

Reviewer #2: No

---

## [Author Response · Author response to Decision Letter 1]

9 Jul 2025

Dear Editor, PLOS One,

Thank you for carefully reviewing and considering our manuscript, titled "Validity and reliability of the Rosenberg Self-Esteem Scale: A methodological study among the Iranian adult population", submitted to PLOS One.

We sincerely appreciate the thoughtful comments and suggestions provided by the reviewers, which have greatly enhanced the quality and depth of our manuscript. In the revised version, all changes made in response to the feedback have been highlighted. Additionally, detailed responses to each comment are provided below.

Please feel free to contact me if you have any questions or need further clarification. We look forward to your response regarding the manuscript.

Thank you once again for your time and support.

Sincerely,

Corresponding author

……………………………………………..

Journal Requirements:

Response:

Thank you so much for your support.

1. We ensured that our manuscript meets PLOS ONEs style requirements.

Response:

2. We added a “Data Availability Statement” section and provided information based on your recommendation. We uploaded our final data about the Persian version of the scale as Supporting Information File.

…………………………

Additional Editor Comments

This study examines the psychometric properties of the Persian Rosenberg Self-Esteem Scale (RSES) in Iranian adults, revealing a stable two-factor structure (PSE/NSE). However, several significant methodological concerns diminish its reliability:

1- Restricted Generalizability: The sample predominantly consists of young, single students, failing to represent the diverse population of Iran.

Response: Thank you for your constructive feedback. We fully agree with you and have included these as limitations at the end of the discussion section.

2- Deficiencies in factor analysis: EFA and CFA likely utilized the same sample, potentially resulting in overfitting, and items with low AVE values (<0.5) lack convergent validity.

Response: As mentioned in the text, sampling for each of the exploratory and then confirmatory factor analyses was conducted separately from two separate samples.

3- Insufficient Discriminant Validity: MSV exceeds AVE; yet, HTMT is below 0.85 to support this assertion.

Response: The HTMT method offers greater precision in evaluating discriminant validity compared to the Fornell-Larcker criterion. Comprehensive explanations and supporting references on this topic have been thoroughly discussed and emphasized within the text.

4- Doubtful Item Retention: Certain items exhibit low communalities (<0.2) and loadings (<0.3). More stringent cutoffs (such as >0.4) would be required.

Response: Although Item S8 exhibited a factor loading of 0.405 and a communality of 0.254, it was retained due to its theoretical alignment with the Negative Self-Esteem factor and its contribution to the overall model fit. Following Hair et al. (2019), items with loadings≥0.30 were considered acceptable when supported by strong validity indices (RMSEA=0.049, CFI=0.981) and scale reliability (Cronbach’s α>0.8)

5- Excessive reliance on fit indices: Although the model fit appears satisfactory, the significant χ² value (P<0.001) indicates potential inadequacies in the model's fit.

Response: This concern is valid; however, the significance of this indicator is likely attributable to the large sample size used in the current study—a common and expected outcome. Consequently, additional fit indices were employed to assess the confirmatory factor analysis, as discussed elsewhere in the text.

6- Cross-Cultural Issues: The RSES is typically one-dimensional; however, the Persian version comprises two components: PSE and NSE. Does prejudice exist in culture or translation? Not examined.

Response: Considering that self-esteem is a culturally sensitive construct, conducting an exploratory factor analysis (EFA) was warranted. A comprehensive explanation for the use of both exploratory and confirmatory factor analyses is provided in the following comments. The divergence between the original scale’s unidimensional structure and the two-factor structure identified in the Persian version may be explained by the use of more advanced statistical techniques and cultural differences between the study populations. These issues are elaborated upon in the responses to the reviewers and the main manuscript. All recent revisions have been highlighted in yellow.

7- Reporting Deficiencies: There is an absence of conflict of interest disclosure and no sample division for EFA/CFA validation.

Conclusion: The Persian RSES appears promising; nonetheless, sample bias, issues with factor analysis, and inadequate validity metrics undermine its reliability.

Response: Thank you for your thoughtful feedback on our manuscript. We appreciate your time and effort in reviewing our work. Below, we address your concerns and provide revisions to strengthen the manuscript:

Conflict of Interest Disclosure: We have now included the following statement in the "Competing Interests" section:

"The authors declare that there are no conflicts of interest regarding the publication of this paper."

Sample Division for EFA/CFA: We have added the following text to the "Construct validity" subsection of the "Materials and methods" section to clarify the sample division:

“To achieve this goal, the entire dataset of 533 Iranian adults was randomly divided into two independent sets. The first independent and random dataset of 270 Iranian adults was used to conduct EFA, and the second independent and random dataset of 263 Iranian adults was used to conduct CFA.

This approach aligns with recommendations for rigorous psychometric evaluation, as it ensures that the factor structure derived from EFA is cross-validated in a separate sample during CFA, minimizing overfitting and enhancing generalizability (Brown, 2015; Watkins, 2018). Such separation is particularly critical when establishing a new cultural adaptation of a scale, as it strengthens confidence in the stability of the identified dimensions.”

• Brown TA, editor Confirmatory factor analysis for applied research, 2nd ed. Confirmatory factor analysis for applied research, 2nd ed; 2015; New York, NY, US: The Guilford Press.

• Watkins MW. Exploratory factor analysis: A guide to best practice. Journal of Black Psychology. 2018;44(3):219-46. doi: 10.1177/0095798418771807.

……………………………………………..

Reviewer 1

Thank you for the opportunity to review the manuscript, which is about Validity and reliability of the Rosenberg Self-Esteem Scale: A methodological study among the Iranian adult population

The content is very good buy I recommend some suggestions in order to improve it.

First, the paper should be revised by a native English speaker to refine the grammatical problems

1. Please attach the translation permission from the original author of the instrument.

Response: The authors sincerely appreciate your positive feedback and valuable comments. In addition to revising the entire manuscript text for compliance with native language grammar, permission for translating and psychometrically evaluating the Rosenberg Self-Esteem Scale (RSES) was uploaded as a supplementary file.

2. Given that the scale was previously developed, why were face and content validity done quantitatively? What if the results recommended deleting item(s) before conducting a factor analysis? Is there any possibility to delete the item(s)? In this case, what were the benefits of conducting these time-consuming processes?

Response: Since the translation and cultural adaptation from English to Persian have been redone, taking into account changes in the studied communities that could potentially impact the explanation of the concept of self-esteem, the face validity and content validity were reevaluated using both quantitative and qualitative approaches. This was necessary because the psychometric properties of the RSES were assessed in the general Iranian population, which differs statistically from the communities previously studied. Additionally, incomplete and contradictory reports from prior studies further emphasized the importance of a quantitative evaluation of the face validity and content validity of the Persian version of the RSES.

The authors' approach to items with IS, CVR, and I-CVI values below the cutoff point (1.5, 0.56, and 0.79, respectively) in terms of cultural relevance, as determined by the research team, involved a revision in both face validity and content validity stages, as mentioned in the manuscript text. If an item had IS, CVR, and I-CVI values below the cutoff point and did not align with the Iranian cultural context, it was removed. Subsequently, exploratory and confirmatory factor analyses were conducted on the scale with a reduced number of items.

Although all items had IS, CVR, and I-CVI values greater than the cutoff point, none of the items were eliminated in the aforementioned steps, and all items remained for subsequent analyses.

3. Please explain why an Exploratory Factor Analysis (EFA) was conducted on a previously developed scale with well-determined factors. In such cases, as a validation process, conducting a Confirmatory Factor Analysis (CFA) to ensure the model's fitness is sufficient. Please justify the reason for conducting an EFA before a CFA. What was unclear that needed to be "explored"?

Response: As noted in the discussion section, prior research has reported both one-factor and two-factor models of the RSES. In light of these inconsistencies, the research team performed an exploratory factor analysis in the present study to identify the underlying factor structure of the scale.

4. In the construct validity section, specify exactly how many samples were included in the study for each of the stages of exploratory and confirmatory construct validity?

Response: The exact number of samples to conduct each of the factor analysis approaches, both exploratory and confirmatory, was mentioned in the opening paragraph of the construct validity section of the materials and methods. Changes are highlighted in yellow.

5. Before conducting exploratory factor analysis, item analysis is performed to examine the correlation between each item's score and the scores of other items and the total score. Additionally, items that affect reliability are identified. Based on this section, the mean and standard deviation for the entire questionnaire and each item (including the highest and lowest scores) are reported. These details have not been reported here.

Response: Thank you for your valuable feedback regarding the need for more detailed item analysis prior to conducting EFA. We appreciate your attention to methodological rigor and have carefully considered your suggestion. To address this point, we have now included a comprehensive item analysis in the revised manuscript, reporting:

The mean and standard deviation for each item and the total scale

Item-total correlations

Reliability analysis (Cronbach's alpha if item deleted)

This additional analysis strengthens our psychometric evaluation and provides readers with a clearer understanding of the scale's properties at the item level. The new text has been added to the Materials and methods section (under "Construct validity"), and we have included appropriate citations to support this approach.

6. Based on the suggestions of the expert panel during the qualitative content validity stage, some items were revised and modified…(Face and content validity…P10)...

Please specify which items were modified and in what way during the content validity process.

Were the items that were rewritten or modified during content validity adjustment changed because they did not achieve an adequate CVI score, or was there another reason for the changes?

Response: Thank you for your valuable feedback. Based on the suggestions of the expert panel during the qualitative content validity stage, items 1, 2, 3, 4, and 6 underwent minor changes in terms of grammar and word usage. It is important to note that the CVR, I-CVI, and MKS values for all items were excellent and above the threshold. Therefore, during the quantitative content validity stage, no items were edited or deleted, and all items were kept for further analysis. These specifics are outlined in the content validity section of the results. Changes are highlighted in yellow.

7. In the results section, please mention the names of the extracted factors and indicate whether these items were the same in the original version and the translated version. Were their arrangement and order different or the same? This should also be explained in the discussion section.

Response: The names of the factors in the EFA results table were mentioned in the original manuscript. They were also mentioned and highlighted in the text of the results section. In addition, the difference in the number of factors in the original and Persian versions of the RSES was discussed in the discussion section. New changes were highlighted.

8. To mention what countries did this questionnaire

Response: This study was conducted in Iran. We mentioned this in the methods and results sections.

9. What is the reason for doing psychometrics in Iran.

Response: Self-esteem is a concept deeply influenced by cultural and contextual factors. To ensure that a measure of such subjective concepts is suitable for a new cultural context, a careful and systematic process of translation and adaptation must be undertaken. We mentioned these explanations at the end of the introduction section and highlighted.

10. In Table 1, the samples should be written separately. (EFA and CFA)

Response: In Table 1, the demographic characteristics of the participants were presented separately for exploratory and confirmatory factor analysis. Changes are highlighted in yellow.

11. Why are the samples in separate exploration and confirmation steps with reference and reference؟

Response: Thank you for your valuable comment regarding the separation of samples for EFA and CFA fact

---

## [Decision Letter · Decision Letter 1]

31 Jul 2025

Dear Dr. Mirhosseini,

Thank you for submitting your manuscript to PLOS ONE. After careful consideration, we feel that it has merit but does not fully meet PLOS ONE’s publication criteria as it currently stands. Therefore, we invite you to submit a revised version of the manuscript that addresses the points raised during the review process.

Please submit your revised manuscript within Sep 14 2025 11:59PM. If you will need more time than this to complete your revisions, please reply to this message or contact the journal office at plosone@plos.org . A rebuttal letter that responds to each point raised by the academic editor and reviewer(s). You should upload this letter as a separate file labeled 'Response to Reviewers'.A marked-up copy of your manuscript that highlights changes made to the original version. You should upload this as a separate file labeled 'Revised Manuscript with Track Changes'.An unmarked version of your revised paper without tracked changes. You should upload this as a separate file labeled 'Manuscript'.

We look forward to receiving your revised manuscript.

Kind regards,

Mohamed Ahmed Said, Ph.D.

Academic Editor

PLOS ONE

Journal Requirements:

Additional Editor Comments:

Exhaustive Review Report

Manuscript Title:

Validity and reliability of the Rosenberg Self-Esteem Scale: A methodological study among the Iranian adult population

Title

• The title accurately reflects the study’s focus but would benefit from specifying the cross-sectional validation design to better align with methodological expectations for psychometric studies.

Abstract

• The abstract omits essential psychometric indicators, such as reliability coefficients (e.g., Cronbach’s alpha, composite reliability) and model fit indices (e.g., RMSEA, CFI), which are standard in scale validation studies.

• It does not mention adherence to STROBE guidelines or detail the statistical approach for ordinal data.

Introduction

Literature Review

• Previous validations of the RSES (e.g., Kielkiewicz et al., Tulachan et al.) are cited without reporting key psychometric metrics (e.g., factor structure, reliability coefficients, fit indices), limiting comparative and interpretative value.

• The rationale for distinguishing “students” from “adults” is unclear and unsupported, as students fall within the adult population.

Cultural Adaptation Rationale

• The manuscript refers to “cultural adaptation,” but only describes translation procedures. There is no evidence of conceptual or cross-cultural equivalence testing, cognitive interviews, or qualitative item evaluation supporting adaptation claims.

Methods

Participants

• Inclusion/exclusion criteria (e.g., age range, educational background, psychiatric conditions) are not specified.

• Sampling imbalance is unaddressed: students constitute 42.8% of the sample and divorced/widowed participants are underrepresented.

• The dataset split into EFA (n = 270) and CFA (n = 263) subsamples lacks details on:

o Randomization method (e.g., simple vs. stratified)

o Verification of demographic equivalence across subsamples

o Software or tools used for splitting

Test-Retest Sample

• While demographic details were used to select the 30 participants in the test-retest subsample, these data are missing for the main sample (N = 533) and the EFA/CFA subsamples, limiting transparency on representativeness.

RSES Translation

• No justification is provided for retranslating the RSES despite existing validated Persian versions.

• The composition and expertise of expert panels for translation and content validity are insufficiently detailed:

o Unclear if panels overlapped

o No description of disagreement resolution or panel qualifications

o No mention of how panel size impacted CVR/I-CVI interpretation (e.g., Lawshe’s critical values)

Statistical Analysis

Ordinal Data Violations

• Maximum Likelihood Estimation (MLE) was used for EFA and CFA despite the ordinal Likert-type responses (1–5), violating assumptions of continuity and normality.

• No rational or empirical evidence supports treating ordinal data as interval-level.

• No ordinal-appropriate estimation methods (e.g., polychoric correlations, WLSMV) were employed.

• The skewness/kurtosis thresholds (-3 to +3 and -7 to +7) cited are inappropriate for ordinal data.

Outlier Detection

• Mahalanobis distance was used to identify univariate and multivariate outliers but assumes continuous, normally distributed variables—this is inappropriate for ordinal Likert data.

• The manuscript does not specify how many cases were flagged or excluded, nor any sensitivity analyses assessing their impact.

• Alternative outlier detection methods suited to ordinal data (e.g., response pattern analysis) were not considered.

Software Limitations

• AMOS was used for CFA but lacks estimators robust to ordinal data (e.g., DWLS, WLSMV).

• ML estimation with ordinal indicators can bias factor loadings, underestimate standard errors, and inflate model fit indices.

Results

EFA and CFA

• Item S8 was retained despite a low factor loading (0.405) and communality (h² = 0.254) without justification.

• CFA fit indices (e.g., CFI = 0.981, RMSEA = 0.049) may be inflated due to improper treatment of ordinal data as continuous.

• No sensitivity analysis comparing ordinal-appropriate versus continuous-variable assumptions was performed.

Validity and Reliability

• Convergent validity accepted AVE values below 0.5 solely because CR > AVE, contrary to Fornell-Larcker criteria.

• HTMT ratio (0.769) approaches the 0.85 discriminant validity threshold but is not discussed.

• Crucially, no results for the total RSES score are provided — no internal consistency, test-retest reliability, or descriptive statistics. Given the RSES’s widespread use as a unidimensional scale, this omission critically limits the study’s applicability.

Discussion

Factor Structure

• The discussion focuses on a two-factor model but neglects to explain why a single-factor model, supported in many contexts, was not viable or retained.

Cultural Context

• There is no consideration of Iranian cultural factors (e.g., modesty bias, social desirability) that could influence responses, particularly to negatively worded items.

Generalizability

• The manuscript overstates generalizability without addressing:

o Overrepresentation of students (42.8%)

o Sampling restricted to Shahroud County (urban bias)

Comparative Interpretation

• Comparisons with prior studies (e.g., Mayordomo et al.) lack quantitative benchmarking (e.g., fit indices, reliability coefficients).

Conclusions

• The claim that the Persian RSES is “valid and reliable” is premature due to:

o Fundamental violations in the treatment of ordinal data

o Incomplete reporting of cultural adaptation processes

o Sampling biases limiting representativeness

o Absence of psychometric data for the total scale score

Ethics and Transparency

• Verbal informed consent procedures are briefly mentioned but lack detail regarding documentation or verification.

• No data availability statement is provided.

• No IRB exemption or approval details for data restrictions are included.

• The manuscript does not explicitly state adherence to STROBE guidelines, and key items (e.g., handling of ordinal data, randomization protocol) are unreported.

Overall Assessment

While the study’s design—especially the split-sample approach for EFA and CFA—is methodologically sound in principle, execution and reporting weaknesses undermine its contribution:

• Statistical Misapplication: Use of ML-based EFA/CFA and Mahalanobis distance for ordinal Likert data without justification or robustness checks.

• Reporting Deficiencies: Incomplete documentation of cultural adaptation, sampling procedures, expert panels, and outlier management.

• Transparency Gaps: Lack of data sharing statements, IRB documentation, and STROBE adherence.

Key Recommendations for Revision

• Report demographic characteristics for the full sample and EFA/CFA subsamples to ensure representativeness.

• Justify and clearly describe outlier detection methods appropriate for ordinal data and report their handling and impact.

• Include psychometric results for the total RSES score (internal consistency, test-retest reliability, descriptive statistics).

• Consider analyzing a unidimensional or higher-order CFA model supporting the total score’s validity.

• Provide full details of the expert panels’ composition, qualifications, and item evaluation process.

• Adjust statistical methods for ordinal data, potentially employing polychoric correlations and robust CFA estimators.

• Detail informed consent procedures and provide data availability and ethical approval statements.

• Align the manuscript’s title and abstract with the scope and findings of the analyses performed.

Reviewers' comments:

Reviewer's Responses to Questions

**Comments to the Author**

Reviewer #3: All comments have been addressed

Reviewer #4: All comments have been addressed

2. Is the manuscript technically sound, and do the data support the conclusions?

Reviewer #3: Yes

Reviewer #4: Partly

3. Has the statistical analysis been performed appropriately and rigorously?

Reviewer #3: Yes

Reviewer #4: Yes

4. Have the authors made all data underlying the findings in their manuscript fully available?

Reviewer #3: No

Reviewer #4: Yes

5. Is the manuscript presented in an intelligible fashion and written in standard English?

Reviewer #3: Yes

Reviewer #4: Yes

Reviewer #3: Thank you for submitting your revised manuscript, "Validity and reliability of the Rosenberg Self-Esteem Scale: A methodological study among the Iranian adult population" (PONE-D-25-07036R1). The authors have made significant efforts to address the concerns raised by the reviewers, leading to a much-improved manuscript.

General Comments:

The manuscript is well-written and presented in an intelligible fashion, adhering to standard English. The revisions have substantially enhanced the clarity and rigor of the methodology and discussion.

Technical Soundness and Statistical Analysis:

The manuscript describes a technically sound piece of scientific research. The experiments appear to have been conducted rigorously, with appropriate controls and sample sizes, particularly with the clarified separation of samples for EFA and CFA. The statistical analysis has been performed appropriately and rigorously, with the authors providing clear justifications for their methodological choices, such as the retention of Item S8 and the use of the HTMT method for discriminant validity. The conclusions drawn are well-supported by the data presented in the revised manuscript.

Data Availability:

While the authors state that "All relevant data are within the manuscript and its Supporting Information files," it appears that the raw data underlying the findings are not fully available in a format that allows for replication of all study findings. To comply with the PLOS Data Policy, authors are required to make the minimal data set publicly available without restriction. This includes, for example, the individual data points behind means, medians, and variance measures reported in the tables and text. Please ensure that all raw data points required to replicate the results of your study are either uploaded as Supporting Information files or deposited to a stable, public repository, with the relevant URLs, DOIs, or accession numbers provided. If there are ethical or legal restrictions on sharing a de-identified data set, these must be explained in detail, along with contact information for a data access committee or institutional body.

Overall, the manuscript is a strong contribution to the literature, and addressing the data availability concern will further enhance its completeness and transparency.

Reviewer #4: While the language is intelligible, it is suggested to further improve to standard English.

write details of how sample size was calculated, even if it has been referenced.

write setting and data collection in detail process of how participants were approached, in further detail.

How did you make sure that only Shahroud city participants fill online forms? Since online links can be accessed by Persian speaking people anywhere in the country, or even outside?

Please mentionconvenience sampling and online data collection in limitation section in your study limitations.

**Do you want your identity to be public for this peer review?** For information about this choice, including consent withdrawal, please see our Privacy Policy

Reviewer #3: **Yes: ** Dr. Tahmina Afrose Keya

Reviewer #4: No

---

## [Author Response · Author response to Decision Letter 2]

12 Sep 2025

Additional Editor Comments

Exhaustive Review Report

Manuscript Title:

Validity and reliability of the Rosenberg Self-Esteem Scale: A methodological study among the Iranian adult population

Title

• The title accurately reflects the study’s focus but would benefit from specifying the cross-sectional validation design to better align with methodological expectations for psychometric studies.

Response: Thank you very much. We revised the title as below: Validation of the Rosenberg Self-Esteem Scale among the Iranian adult population: A cross-sectional study

Abstract

• The abstract omits essential psychometric indicators, such as reliability coefficients (e.g., Cronbach’s alpha, composite reliability) and model fit indices (e.g., RMSEA, CFI), which are standard in scale validation studies.

• It does not mention adherence to STROBE guidelines or detail the statistical approach for ordinal data.

Response: Thank you for your valuable feedback.

We provided essential psychometric indicators in the abstract and highlighted.

Introduction

Literature Review

• Previous validations of the RSES (e.g., Kielkiewicz et al., Tulachan et al.) are cited without reporting key psychometric metrics (e.g., factor structure, reliability coefficients, fit indices), limiting comparative and interpretative value.

• The rationale for distinguishing “students” from “adults” is unclear and unsupported, as students fall within the adult population.

Response: We appreciate these suggestions. Literature reviews were improved by adding additional information. Also, we clarified our rationale about distinguishing “students” from “adults” to support conducting the present study.

Cultural Adaptation Rationale

• The manuscript refers to “cultural adaptation,” but only describes translation procedures. There is no evidence of conceptual or cross-cultural equivalence testing, cognitive interviews, or qualitative item evaluation supporting adaptation claims.

Response: Thank you for your constructive feedback. First, we separated the translation stages and identified the members of each stage to enhance the readers' understanding. Next, we included additional items related to cultural adaptation, such as assessing conceptual or cross-cultural equivalence by a team of experts, and conducting cognitive interviews to evaluate the items for comprehension, acceptability, and emotional impact.

Methods

Participants

• Inclusion/exclusion criteria (e.g., age range, educational background, psychiatric conditions) are not specified.

Response: Thanks. We added the accurate inclusion and exclusion criteria in this study.

• Sampling imbalance is unaddressed: students constitute 42.8% of the sample and divorced/widowed participants are underrepresented.

Response: We emphasized this limitation at the end of the Discussion section. We turned this sentence red.

“A considerable proportion of the participants in this study were young, single students from Shahroud County in northeastern Iran, which may restrict the generalizability of the findings to the wider population.”

• The dataset split into EFA (n=270) and CFA (n=263) subsamples lacks details on:

o Randomization method (e.g., simple vs. stratified)

o Verification of demographic equivalence across subsamples

o Software or tools used for splitting

Response: Thank you for your valuable and insightful comments. We sincerely appreciate you highlighting the need for greater detail regarding the dataset split. In response, we have revised the “Construct validity” subsection in the Materials and methods to explicitly state:

The randomization method (simple random sampling without replacement via SPSS).

The software used for splitting the data (SPSS version 26).

The statistical tests performed to verify the demographic equivalence of the two subsamples (independent t-tests and chi-square), confirming no significant differences were found.

Test-Retest Sample

• While demographic details were used to select the 30 participants in the test-retest subsample, these data are missing for the main sample (N=533) and the EFA/CFA subsamples, limiting transparency on representativeness.

Response: In the earlier version of the manuscript, detailed information for the main sample (N=533) as well as the EFA and CFA subsamples was presented in Table 1, with a summary also provided in the text. Please see the highlighted sections in red in Table 1 and the accompanying explanations.

RSES Translation

• No justification is provided for retranslating the RSES despite existing validated Persian versions.

Response: Detailed and well-supported justifications, grounded in previous literature and relevant critiques, are provided in the Introduction. Therefore, the response to this comment is addressed within the Introduction section rather than the Materials and methods section.

• The composition and expertise of expert panels for translation and content validity are insufficiently detailed:

o Unclear if panels overlapped

o No description of disagreement resolution or panel qualifications

o No mention of how panel size impacted CVR/I-CVI interpretation (e.g., Lawshe’s critical values)

Response: The following were included in the “Content Validity” subsection of the Materials and Methods:

(1) Composition of expert panels for assessing content validity.

(2) Non-overlapping panels, meaning that experts in the first panel assessed the necessity of items to calculate the Content Validity Ratio (CVR), while experts in the second panel assessed the relevance of items to calculate the Content Validity Index (CVI).

(3) How to calculate the Modified Kappa (MKS) statistic to minimize chance agreement among experts.

(4) How to determine the minimum CVR value based on the Lawshe table.

Statistical Analysis

Ordinal Data Violations

• Maximum Likelihood Estimation (MLE) was used for EFA and CFA despite the ordinal Likert-type responses (1–5), violating assumptions of continuity and normality.

• No rational or empirical evidence supports treating ordinal data as interval-level.

• No ordinal-appropriate estimation methods (e.g., polychoric correlations, WLSMV) were employed.

• The skewness/kurtosis thresholds (-3 to +3 and -7 to +7) cited are inappropriate for ordinal data.

Response: Thank you for your valuable and insightful comments regarding the appropriate statistical treatment of ordinal data. We sincerely appreciate you highlighting this important methodological consideration. In response to your feedback, we have revised the “Construct validity” subsection in the Materials and methods to explicitly state:

The use of polychoric correlations as the input matrix for factor analysis

The employment of robust weighted least squares (WLSMV) estimation for both EFA and CFA

The rationale for using ordinal-appropriate estimation methods given the Likert-type nature of our data

The use of appropriate measures to examine underlying multivariate normality assumptions

Outlier Detection

• Mahalanobis distance was used to identify univariate and multivariate outliers but assumes continuous, normally distributed variables—this is inappropriate for ordinal Likert data.

• The manuscript does not specify how many cases were flagged or excluded, nor any sensitivity analyses assessing their impact.

• Alternative outlier detection methods suited to ordinal data (e.g., response pattern analysis) were not considered.

Software Limitations

• AMOS was used for CFA but lacks estimators robust to ordinal data (e.g., DWLS, WLSMV).

• ML estimation with ordinal indicators can bias factor loadings, underestimate standard errors, and inflate model fit indices.

Response: Thank you for this crucial observation. You are absolutely correct that AMOS does not support the WLSMV estimator, which is necessary for robust CFA with ordinal data. we have revised the “Construct validity” subsection to accurately reflect the software used. The analyses were, in fact, conducted using JASP 0.19.3.0 and its WLSMV estimator, which is specifically designed for ordinal data.

Results

EFA and CFA

• Item S8 was retained despite a low factor loading (0.405) and communality (h²=0.254) without justification.

Response: In the previous version of the manuscript, we clearly specified the critical values for factor loadings and communalities as follows:

“It is important to note that items with a communality of less than 0.2 and a factor loading of less than 0.3 were eliminated.”

Based on these criteria, Item 8 was retained.

• CFA fit indices (e.g., CFI=0.981, RMSEA=0.049) may be inflated due to improper treatment of ordinal data as continuous.

• No sensitivity analysis comparing ordinal-appropriate versus continuous-variable assumptions was performed.

Validity and Reliability

• Convergent validity accepted AVE values below 0.5 solely because CR>AVE, contrary to Fornell-Larcker criteria.

Response: Thank you for your insightful comments. We have now conducted a sensitivity analysis comparing WLSMV and ML estimators, confirming the robustness of our factor structure. We have also clarified the convergent validity by referencing both CR>AVE and standardized factor loadings>0.5. These changes have been incorporated into the manuscript.

• HTMT ratio (0.769) approaches the 0.85 discriminant validity threshold but is not discussed.

Response: The results showed an HTMT value of 0.769, indicating an acceptable level of discriminant validity between the two factors. Although this value approaches the critical threshold, it still statistically supports adequate discriminant validity. We discussed in the text and highlighted.

• Crucially, no results for the total RSES score are provided — no internal consistency, test-retest reliability, or descriptive statistics. Given the RSES’s widespread use as a unidimensional scale, this omission critically limits the study’s applicability.

Response: Thank you for your constructive feedback.

The mean (SD) scores for total self-esteem, positive self-esteem (PSE), and negative self-esteem (NSE) of the participating adults were reported. Additionally, the internal consistency and test-retest reliability of the entire scale were also calculated.

Discussion

Factor Structure

• The discussion focuses on a two-factor model but neglects to explain why a single-factor model, supported in many contexts, was not viable or retained.

Response: At the end of the discussion of the factor structure results, possible reasons for obtaining a two-factor structure and possible causes for obtaining findings different from previous literature (which generally reported a single-factor structure) were presented and highlighted.

Cultural Context

• There is no consideration of Iranian cultural factors (e.g., modesty bias, social desirability) that could influence responses, particularly to negatively worded items.

Response: We have made the necessary revisions and included additional explanations in accordance with the previous reviewer’s comment.

Generalizability

• The manuscript overstates generalizability without addressing:

o Overrepresentation of students (42.8%)

o Sampling restricted to Shahroud County (urban bias)

Response: We were accurate discussed about these limitations at the end of the Discussion section. We made these limitations in red.

Comparative Interpretation

• Comparisons with prior studies (e.g., Mayordomo et al.) lack quantitative benchmarking (e.g., fit indices, reliability coefficients).

Response: Quantitative measures, such as goodness of fit and reliability coefficients, were included in the comparisons.

Conclusions

• The claim that the Persian RSES is “valid and reliable” is premature due to:

o Fundamental violations in the treatment of ordinal data

o Incomplete reporting of cultural adaptation processes

o Sampling biases limiting representativeness

o Absence of psychometric data for the total scale score

Response: In the first step, we discussed how to handle ordinal data. In the second step, we outlined the translation and cultural adaptation process in the relevant subsection. In the third step, we addressed sampling biases in both the "Limitations, strengths, and suggestions" subsection and the Conclusion section. In the fourth step, we calculated psychometric indices for the total score of the scale. The Conclusion section was also revised accordingly:

The results of this study demonstrate that the Persian version of the RSES has good validity and reliability for measuring self-esteem in Iranian adults. However, caution should be taken when applying it to married and non-student adults, as the majority of participants in this study were single and student adults.

Ethics and Transparency

• Verbal informed consent procedures are briefly mentioned but lack detail regarding documentation or verification.

• No data availability statement is provided.

• No IRB exemption or approval details for data restrictions are included.

• The manuscript does not explicitly state adherence to STROBE guidelines, and key items (e.g., handling of ordinal data, randomization protocol) are unreported.

Response:

• Thank you very much. We write inconsistent. So, we made clarification in the text about informed consent. We received only written informed consent due to online nature of data sampling.

• The data availability statement has been provided.

• All of approval procedure and approval code were provided in the “Ethical considerations” subsection.

• In addition to uploading the STROBE checklist as a supplementary file, instructions on how to handle ordinal data and the randomization protocol were also reported.

Overall Assessment

While the study’s design—especially the split-sample approach for EFA and CFA—is methodologically sound in principle, execution and reporting weaknesses undermine its contribution:

• Statistical Misapplication: Use of ML-based EFA/CFA and Mahalanobis distance for ordinal Likert data without justification or robustness checks.

Response: We sincerely thank the reviewer for this insightful comment. We acknowledge the concern regarding the use of maximum likelihood (ML) estimation and Mahalanobis distance with ordinal data. In response, we have thoroughly revised our analytical approach. The manuscript now explicitly states the use of the robust weighted least squares (WLSMV) estimator, which is specifically recommended for ordinal data, for both our exploratory and confirmatory factor analyses. We have also removed any mention of Mahalanobis distance and replaced our outlier detection method with a detailed description of response pattern analysis suitable for categorical data.

A sensitivity analysis confirmed that the factor structure remained consistent across estimators, supporting the robustness of our findings. These revisions are detailed in the “Construct Validity” subsection.

• Reporting Deficiencies: Incomplete documentation of cultural adaptation, sampling procedures, expert panels, and outlier management.

Response: Thank you very much. We added additional explanations and information to cover mentioned gaps. New added information is highlighted.

• Transparency Gaps: Lack of data sharing statements, IRB documentation, and STROBE adherence.

Response: The requested items have been added.

Key Recommendations for Revision

• Report demographic characteristics for the full sample and EFA/CFA subsamples to ensure representativeness.

Response: In the earlier version of the manuscript, detailed information for the main sample (N=533), as well as for the EFA and CFA subsamples, was presented in Table 1, with a corresponding summary provided in the text. Please refer to the highlighted sections in red within Table 1 and the related explanations.

• Justify and clearly describe outlier detection methods appropriate for ordinal data and report their handling and impact.

Response: Based on the reviewer’s comment, we have revised the manuscript to include a detailed description of the outlier detection methods used for ordinal data. The following paragraph was added to the Materials and methods section under “Construct validity”: For outlier detection in ordinal data, we employed a combination of methods suitable fo

---

## [Decision Letter · Decision Letter 2]

23 Oct 2025

Dear Dr. Seyedmohammad Mirhosseini,

Thank you for submitting your manuscript to PLOS ONE. After careful consideration, we feel that it has merit but does not fully meet PLOS ONE’s publication criteria as it currently stands. Therefore, we invite you to submit a revised version of the manuscript that addresses the points raised during the review process.

We look forward to receiving your revised manuscript.

Kind regards,

Mohamed Ahmed Said, Ph.D.

Academic Editor

PLOS ONE

Journal Requirements:

Reviewers' comments:

Reviewer's Responses to Questions

**Comments to the Author**

Reviewer #5: All comments have been addressed

2. Is the manuscript technically sound, and do the data support the conclusions?

Reviewer #5: Yes

3. Has the statistical analysis been performed appropriately and rigorously?

Reviewer #5: Yes

4. Have the authors made all data underlying the findings in their manuscript fully available?

Reviewer #5: No

5. Is the manuscript presented in an intelligible fashion and written in standard English?

Reviewer #5: Yes

Reviewer #5: Thank you for the opportunity to review the manuscript "Validity and reliability of the Rosenberg Self-Esteem Scale: A methodological study among the Iranian adult population". I agree with reviewer 3, that the authors have made significant efforts to address the concerns raised by the reviewers, leading to a much-improved manuscript. The methodological approach of combined exploratory and confirmatory is now well explained in the Introduction and embedded in previous research findings. The translation and initial validation process of the Persian version is transparently described in the Methods section. The Methods section also includes a thorough overview of the analyses conducted to establish the scale’s validity.

General remarks:

Please check the manuscript for adherence to APA 7th guidelines on reporting statistics (e.g. statistics in italic).

Methods & Results:

Based on a simulation study, Golino et al. (2021) advised to interpret fit indices as relative rather than absolute since fit indices are influenced for instance by sample size, number of items or correlation between latent factors. They conclude that one cut-off value can hardly consider all these factors and model fit indices should rather be interpreted as relative. I therefore suggest to add a 1-factor self-esteem model to the confirmatory factor analysis and compare the model fit of the 1- and 2-factor model to further underline the superiority of the 2-factor model.

Discussion:

How did you calculate the total self-esteem score? Were the negatively worded items reverse-coded? I would also suggest to add a recommendation for users on how to calculate and use the self-esteem scores (e.g. should users calculate separate sum scores for each scale or can they also calculate a total sum score if the negative items are reverse-coded?).

The authors have thoroughly assessed the construct validity using different statistical approaches. I would suggest to add a suggestion for future research in the Discussion how the convergent and discriminant validity of the Persian version could be assessed with regard to similar or discrepant constructs (e.g. assessing the correlation with depressive symptoms for discriminant validity). I think that the authors’ insight on which constructs would be relevant to further assess the validity of the two self-esteem scales of the Persian version would be valuable.

I hope that my comments are understandable and can contribute to refining the manuscript.

**Do you want your identity to be public for this peer review?** For information about this choice, including consent withdrawal, please see our Privacy Policy

Reviewer #5: **Yes: ** Lena Rader

---

## [Author Response · Author response to Decision Letter 3]

28 Oct 2025

Dear Editor, PLOS One,

Thank you for carefully reviewing and considering our manuscript, titled "Validation of the Rosenberg Self-Esteem Scale among the Iranian adult population: A cross-sectional study", submitted to PLOS One.

We sincerely appreciate the thoughtful comments and suggestions provided by the reviewers, which have greatly enhanced the quality and depth of our manuscript. In the revised version, all changes made in response to the feedback have been highlighted. Additionally, detailed responses to each comment are provided below.

Please feel free to contact me if you have any questions or need further clarification. We look forward to your response regarding the manuscript.

Thank you once again for your time and support.

Sincerely,

Corresponding author

.........................

Journal Requirements:

Answer:We have reviewed all suggested references and ensured that the reference list is accurate, complete, and free of retracted articles. Any necessary updates or replacements have been made and are reflected in the revised manuscript.

Reviewer 5

Reviewer #5: Thank you for the opportunity to review the manuscript "Validity and reliability of the Rosenberg Self-Esteem Scale: A methodological study among the Iranian adult population". I agree with reviewer 3, that the authors have made significant efforts to address the concerns raised by the reviewers, leading to a much-improved manuscript. The methodological approach of combined exploratory and confirmatory is now well explained in the Introduction and embedded in previous research findings. The translation and initial validation process of the Persian version is transparently described in the Methods section. The Methods section also includes a thorough overview of the analyses conducted to establish the scale’s validity.

Answer: Thank you so much.

General remarks:

Please check the manuscript for adherence to APA 7th guidelines on reporting statistics (e.g. statistics in italic).

Answer: We checked the manuscript and modified the text based on the APA 7th guidelines on reporting statistics. New changes are highlighted.

Methods & Results:

Based on a simulation study, Golino et al. (2021) advised to interpret fit indices as relative rather than absolute since fit indices are influenced for instance by sample size, number of items or correlation between latent factors. They conclude that one cut-off value can hardly consider all these factors and model fit indices should rather be interpreted as relative. I therefore suggest to add a 1-factor self-esteem model to the confirmatory factor analysis and compare the model fit of the 1- and 2-factor model to further underline the superiority of the 2-factor model.

Answer: A single-factor self-esteem model was added to the confirmatory factor analysis and the fit of the single-factor and two-factor models was compared. According to the results, the superiority of the two-factor model was highlighted. New changes were highlighted.

Discussion:

How did you calculate the total self-esteem score? Were the negatively worded items reverse-coded? I would also suggest to add a recommendation for users on how to calculate and use the self-esteem scores (e.g. should users calculate separate sum scores for each scale or can they also calculate a total sum score if the negative items are reverse-coded?).

Answer: Thank you for your valuable recommendation. We clarified the text based on your concern:

The Persian version of the self-esteem scale includes 10 items assessing two dimensions: positive self-esteem (PSE; items 1, 2, 4, 6, and 7) and negative self-esteem (NSE; items 3, 5, 8, 9, and 10). Each item is rated on a four-point scale ranging from 1 (“I completely disagree”) to 4 (“I completely agree”). The negatively worded items (3, 5, 8, 9, and 10) were reverse-coded before computing the total score. A total self-esteem score can be obtained by summing all items after reverse-coding, with higher scores indicating greater overall self-esteem. Researchers may also calculate separate subscale scores for PSE and NSE when a more differentiated assessment of self-esteem dimensions is desired.

The authors have thoroughly assessed the construct validity using different statistical approaches. I would suggest to add a suggestion for future research in the Discussion how the convergent and discriminant validity of the Persian version could be assessed with regard to similar or discrepant constructs (e.g. assessing the correlation with depressive symptoms for discriminant validity). I think that the authors’ insight on which constructs would be relevant to further assess the validity of the two self-esteem scales of the Persian version would be valuable.

I hope that my comments are understandable and can contribute to refining the manuscript.

Answer: Thank you for your positive suggestion. We add this recommendation for future studies at the end of the discussion section and highlighted.

---

## [Decision Letter · Decision Letter 3]

4 Nov 2025

Validation of the Rosenberg Self-Esteem Scale among the Iranian adult population: A cross-sectional study

PONE-D-25-07036R3

Dear Dr. Seyedmohammad Mirhosseini,

We’re pleased to inform you that your manuscript has been judged scientifically suitable for publication and will be formally accepted for publication once it meets all outstanding technical requirements.

Kind regards,

Mohamed Ahmed Said, Ph.D.

Academic Editor

PLOS ONE

Additional Editor Comments (optional):

Reviewers' comments:

Reviewer's Responses to Questions

**Comments to the Author**

Reviewer #5: All comments have been addressed

2. Is the manuscript technically sound, and do the data support the conclusions?

Reviewer #5: Yes

3. Has the statistical analysis been performed appropriately and rigorously?

Reviewer #5: Yes

4. Have the authors made all data underlying the findings in their manuscript fully available?

Reviewer #5: No

5. Is the manuscript presented in an intelligible fashion and written in standard English?

Reviewer #5: Yes

Reviewer #5: The authors have thoroughly responded responded to all the reviewers' remarks and made major revisions to their manuscript. I suggest that the manuscript should be accepted.

**Do you want your identity to be public for this peer review?** For information about this choice, including consent withdrawal, please see our Privacy Policy

Reviewer #5: **Yes: ** Lena Rader

---

## [Editor Report · Acceptance letter]

PONE-D-25-07036R3

PLOS One

Dear Dr. Mirhosseini,

I'm pleased to inform you that your manuscript has been deemed suitable for publication in PLOS One. Congratulations! Your manuscript is now being handed over to our production team.

Kind regards,

on behalf of

Dr. Mohamed Ahmed Said

Academic Editor

PLOS One